# Precision Medicine to Treat Urothelial Carcinoma—The Way Forward

**DOI:** 10.3390/cancers15113024

**Published:** 2023-06-01

**Authors:** Carvy Floyd Luceno, Won Jin Jeon, Ravand Samaeekia, John Shin, Guru P. Sonpavde

**Affiliations:** 1Department of Internal Medicine, Loma Linda University Medical Center, Loma Linda, CA 92354, USA; 2Department of Medical Oncology/Hematology, Loma Linda University Medical Center, Loma Linda, CA 92354, USA; 3Department of Medical Oncology, Section of Genitourinary Oncology and Phase I Clinical Research, AdventHealth Cancer Institute, University of Central Florida, Orlando, FL 32816, USA

**Keywords:** urothelial carcinoma, targeted therapy, precision medicine, immunotherapy, biomarkers

## Abstract

**Simple Summary:**

The treatment of urothelial carcinoma is challenging. While known therapies are effective, the results can be variable. Known treatment modalities include transurethral resection of the bladder tumor, intravesicular BCG, chemotherapy, immune check point inhibitors, and antibody drug conjugates. Finding certain patient and tumor characteristics to determine responders to therapy can help personalize the treatment approach and optimize results. Precision medicine offers a solution to this problem by providing clinicians with tools such as liquid biopsies, prognostication models, and biomarkers to identify essential patient characteristics. Furthermore, precision medicine improves treatment efficacy by identifying and exploiting specific targets. In this review, we discuss available tools in precision medicine, describe ongoing clinical trials, and identify areas for future study.

**Abstract:**

The treatment of urothelial carcinoma (UC) is challenging given its molecular heterogeneity and variable response to current therapies. To address this, many tools, including tumor biomarker assessment and liquid biopsies, have been developed to predict prognosis and treatment response. Approved therapeutic modalities for UC currently include chemotherapy, immune checkpoint inhibitors, receptor tyrosine kinase inhibitors, and antibody drug conjugates. Ongoing investigations to improve the treatment of UC include the search for actionable alterations and the testing of novel therapies. An important objective in recent studies has been to increase efficacy while decreasing toxicity by taking into account unique patient and tumor-related factors—an endeavor called precision medicine. The aim of this review is to highlight advancements in the treatment of UC, describe ongoing clinical trials, and identify areas for future study in the context of precision medicine.

## 1. Introduction

Since the advent of the Precision Medicine Initiative in 2015, efforts have been made to develop personalized diagnostic and therapeutic approaches to cancer care that take into account individual differences including somatic genomic, transcriptomic, and proteomic alterations, as well as germline genetic mutations. Significant advancements have been made that have changed the therapeutic landscape of advanced urothelial carcinoma, including the emergence of immune checkpoint inhibitors (PD1/L1 inhibitors), antibody drug conjugates (enfortumab vedotin, sacituzumab govitecan), and a targeted agent (erdafitinib). Moreover, the application of PD1/L1 inhibitors to earlier disease settings has improved outcomes, e.g., muscle-invasive and non-muscle invasive bladder.

The development of novel diagnostic modalities, molecular data, and therapeutic approaches have led to important strides in precision medicine in the management of this disease. Unfortunately, PD-L1 immunohistochemistry has performed poorly in randomized phase III trials of advanced disease, which have led to withdrawal of approvals of first-line atezolizumab and pembrolizumab for PD-L1 high-expressing advanced cisplatin-ineligible urothelial carcinoma. In this review, we summarize the impact of novel emerging biomarkers and precision medicine on urothelial carcinoma and the developments on the horizon that may enable greater personalization of therapy for this disease (Figure 1). We explore how biomarkers may imply favorable treatment outcomes for patients with MIBC and locally advanced or metastatic UC. The possibility of selecting patients who should receive neoadjuvant or adjuvant therapies according to liquid biopsies, prognostic tools, and biomarkers is also discussed.

## 2. Better Understanding of Tumor Biology and Emergence of Liquid Biopsies

The inter- and intra-tumor heterogeneity of bladder cancer has been well documented and is a challenge in delivering precise and efficacious treatment. “Liquid biopsies” which utilize peripheral blood to assess for circulating tumor cells (CTCs) and circulating tumor DNA (ctDNA) have gained significant attention due to their non-invasive nature and potential in identifying driver mutations. CTCs are thought to originate in the primary tumor and enter circulation as they metastasize to distant organs, while ctDNA includes DNA mutations, epigenetic alterations, and other forms of tumor-specific abnormalities [1,2]. Vandekerkhove et al. demonstrated a mutational concordance of 83.4% between ctDNA and matched bladder tumor tissue [3]. In addition, 90% of mutations remained consistent across serial ctDNA samples, while concordance for serial tumor tissue was significantly lower, indicating that the identification of driver mutations might be better identified in plasma.

CTCs have also shown potential in predicting therapeutic response and prognosis. Soave et al. demonstrated that the presence of CTCs is associated with increased disease recurrence, cancer-specific mortality, and overall mortality in patients who did not receive adjuvant chemotherapy prior to radical cystectomy in patients with recurrent Ta, T1, CIS refractory to TURBT, or muscle-invasive bladder cancer (MIBC) [4]. In patients with CTCs who received adjuvant chemotherapy, there was no difference in disease recurrence and cancer-specific or overall mortality. The presence of CTCs may provide guidance on whether to treat UC with adjuvant chemotherapy; however, further investigation is required before this can be clinically applied. Another study demonstrated that quantifying CTCs may also predict prognosis. Alva et al. collected blood samples from 20 patients with BC who were eligible for neoadjuvant chemotherapy and observed that patients with medium to high tumor cell levels at baseline and follow-up had an unfavorable pathological stage disease [5].

Liquid biopsies have also shown potential in identifying minimal residual disease (MRD) and predicting response to ICIs. A retrospective analysis of the phase 3 IMvigor010 study that evaluated adjuvant treatment with atezolizumab compared with observation in muscle-invasive urothelial carcinoma revealed that ctDNA using a tumor-informed plasma cell-free DNA profiling platform may provide information on therapeutic response and prognosis [6]. The original trial did not meet its primary endpoint of improved disease-free survival and did not support the use of atezolizumab as an adjuvant chemotherapy [7]. However, the subsequent study revealed a prevalence of 37% for ctDNA positivity post-surgically and found that these patients showed a substantial disease-free survival (DFS) (HR = 0.58 [95% CI 0.43–0.79]; *p* = 0.0005) and overall survival (OS) (HR = 0.59 [95% CI 0.41–0.86]; *p* = 0.0059) with atezolizumab vs. obs [8]. In addition, ctDNA-positive patients who also had high levels of PD-L1 and TMB were found to have an additional survival benefit with a hazard ratio (HR) of 0.52 ([95% CI 0.331–0.82]; *p* = 0.004) and 0.34 ([95% CI 0.19–0.6]; *p* < 0.0001]), respectively [9]. The rate of ctDNA clearance from C1D1 to C3D1 was noted to be higher in atezolizumab (18.2%) vs. observation (3.8%) (*p* = 0.0048) and was also associated with improved DFS and OS with HRs of 0.26 (95% CI: 0.12, 0.56) and 0.41 (95% CI: 0.10, 1.70), respectively. Similarly, ctDNA dynamics in the neoadjuvant setting appeared to predict outcomes and may help guide therapy, especially bladder preservation. These findings suggest that the detection of ctDNA in the peri-operative setting might guide tailored therapy.

Other non-invasive molecular platforms are also emerging. Cell-free methylated DNA may complement the ability of genomic alterations to detect MRD and may enable a more user-friendly tumor-non-informed off-the-shelf ctDNA platform to identify MRD [10]. Urinary tumor (ut)-DNA appeared promising as a surrogate for MRD in both MIBC and non-MIBC (NMIBC) [11,12]. Indeed, a combination of ctDNA and utDNA dynamics correlated significantly with pathologic response in MIBC, suggesting that this platform may help select patients for bladder-preserving treatment [12,13]. Interestingly, utDNA could be used to infer TMB and advance patient selection for immunotherapy [14].

## 3. Immune Checkpoint Inhibitors and Prediction of Therapeutic Response

Precision medicine has shown promise in predicting therapeutic response to immune checkpoint inhibitors (ICIs). One of the mechanisms that allows cancers to evade immune detection is through “immune checkpoints” that are present on the cell surface and result in T-cell suppression. The FDA has approved several ICIs including pembrolizumab, nivolumab, and avelumab that exploit this mechanism in patients with locally advanced or metastatic urothelial carcinoma (mUC) that progresses during or after platinum-based chemotherapy [15,16,17]. In addition to second-line therapy, avelumab is also approved for maintenance therapy for patients with UC that has not progressed on first-line platinum-based chemotherapy [18].

Unfortunately, responses are not universal, and various attempts have been made to find biomarkers that predict outcomes and response to ICIs. Readily available clinical factors have been associated with outcomes. A five-factor prognostic model consisting of clinical and laboratory factors has been shown to predict survival in post-platinum patients with progressive mUC receiving a PD-L1 inhibitor (atezolizumab, avelumab, and durvalumab) [19]. This model takes into account Eastern Cooperative Oncology group-performance status (ECOG-PS), liver metastasis, platelet count, neutrophil/lymphocyte ratio (NLR), and lactate dehydrogenase (LDH) and assigns patients a risk category. This prognostic model may assist in the risk stratification, interpretation, and design of trials incorporating PD-1/PD-L1 inhibitors in the post-platinum progressive disease setting. However, there is a need to identify predictive factors for benefit and response. While PD-L1 protein expression has generally under-performed and exhibited inconsistency in advanced disease, adjuvant nivolumab following surgery for high-risk muscle-invasive disease was approved in Europe for those with PD-L1 high-expressing tumors only based on the more robust improvement of disease-free survival (DFS) in this group in the phase III CheckMate-274 trial. In contrast, the US FDA approved adjuvant nivolumab regardless of PD-L1 expression.

Genomic and transcriptomic panels have all been variably associated with activity of ICIs. Tumor mutation burden (TMB) has shown potential in predicting therapeutic response to ICIs (Table 1). TMB is defined as the number of somatic mutations per megabase of interrogated genomic sequence and has been studied as a predictive biomarker for the response of ICI treatment [20]. IMvigor211 was a study of atezolizumab vs. chemotherapy in platinum-treated mUC that did not meet its primary endpoint of OS in PD-L1-selected patients [21]. However, in the TMB-high subgroup, median OS was longer with atezolizumab (HR 0.68, CI 95% 0.51–0.90). In contrast, the TMB-low tumors had a variety of responses, including complete and partial responses as well as prolonged OS. CheckMate-275 was a single-arm phase II trial evaluating nivolumab as a monotherapy for metastatic or surgically unresectable advanced UC and progression or recurrence after at least one platinum-based chemotherapy regimen [16]. Similar discoveries were made when CheckMate-275 was analyzed again after 3 years of follow-up to investigate TMB as a predictor of ICI response. They found TMB alone showed a positive association with overall response rate (ORR) with nivolumab [OR (95% CI): 2.13 (1.26–3.60), *p* < 0.05] regardless of baseline tumor PD-L1 expression, although the tail of the curve showed a longer survival benefit in patients with both high TMB and PD-L1 expression ≥1% [22]. Archival tumor molecular profiling of patients enrolled in the phase III first-line maintenance avelumab trial revealed that survival extension by avelumab was associated with PD-L1 expression by tumor cells, TMB, APOBEC mutation signatures, expression of innate and adaptive immune activity genes, and the number of alleles encoding high-affinity Fcꝩ receptors [23]. In contrast, pathways connected to growth and angiogenesis appeared to be associated with reduced survival benefit. The combination of TMB and PD-L1 status was a better predictor of OS than PD-L1 alone (*p* = 0.013) in another retrospective study from a single institution. Nassar et al. identified that a composite panel of clinical, laboratory, and genomic data including TMB/SNV count, visceral metastases, and neutrophil/lymphocyte ratio (NLR) highly correlated with tumor regression [AUC (95% CI) = 0.90 (0.80, 0.99)] [24]. While FGFR activating alterations were associated with poor activity of ICIs in one study [25], there is contrasting evidence based on potentially favorable stromal factors in tumors harboring FGFR-activating genomic alterations [22]. Notably, tumor-agnostic approval of pembrolizumab has been conferred to treat advanced solid tumors with high TMB or microsatellite instability (MSI) [26,27].

Goswami et al. further analyzed two separate cohorts of patients in CheckMate-275 and in IMvigor210 who received nivolumab and atezolizumab, respectively, for progressive disease for AT-rich interactive domain-containing protein 1A (ARID1A) mutations and immune cytokine CXCL13 gene expression in baseline tumor tissues [28]. They interrogated CXCL13 plus ARID1A as a combination biomarker and found that expression may improve prediction of PD1/L1 inhibitor response. In CheckMate-275, patients with ARID1A mutation and high CXCL13 had a median progression free survival (PFS) of 3.7 months (85% CI, 1.8 to NA) and OS of 19.1 months (95% CI, 6.1 to NA) compared to patients with neither alteration who had a median PFS of 1.9 months (95% CI, 1.7–2.0) and median OS of 5.3 months (95% CI, 3.6 to 11.4). IMvigor210 observed that patients with expression of both the ARID1A mutation plus baseline CXCL13 expression had a median OS of 17.8 months (95% CI, 10.4 to NA) compared to 7.1 months (95% CI, 5.5 to 9.9) in patients with no ARID1A mutation and low CXCL13. Analysis of the hazard curves in both studies for CXCL13 and ARID1A mutation status showed positive association of ARID1A mutation and OS with an increase in CXCL13 gene expression. DNA damage response (DDR) gene alterations have also been found to be independently associated with response to PD-1/L1 blockade in patients with mUC. Overall, DDR alteration was associated with a higher response rate (67.9% v 18.8%; *p* < 0.001) in those with likely deleterious DDR alterations (80%) compared with DDR alterations of unknown significance (54%), and in those with wild-type DDR genes (19%; *p* < 0.001) [29].

In CheckMate-275, UC was categorized into molecular subtypes (luminal 1, luminal 2, basal 1, and basal 2), according to the cancer genome atlas. Basal 1 (cluster 1) and luminal 2 (cluster 2) were found to have higher complete response (CR) rates to nivolumab compared to the other subtypes (Basal 2 CR, 0%; luminal 1 CR,1.5%; and luminal 2 CR, 1.8%). This study also evaluated immune gene signature expressions and found that patients with a high interferon-γ signature were more likely to respond to nivolumab than those with a low expression (*p* = 0.0003). Similarly, gene expression molecular subtypes were associated with certain outcomes with neoadjuvant cisplatin-based chemotherapy in a study [30]. Luminal tumors exhibited the best survival and claudin-low tumors were associated with poor survival, regardless of neoadjuvant chemotherapy. Basal tumors showed the most improvement in survival with neoadjuvant chemotherapy compared with surgery alone. In contrast, another study reported that luminal-like subtypes appear more responsive to cisplatin-based neoadjuvant chemotherapy [31]. These authors proposed that a second-generation of subtype-specific biomarkers such as SPP1 (which codes for osteopontin) may facilitate the development of precision neoadjuvant chemotherapy in MIBC.

Gene expression signatures have also been investigated to assess the efficacy of neoadjuvant ICI therapy (Table 2). The ABACUS study investigated neoadjuvant atezolizumab in patients with MIBC and achieved their primary endpoint with a pathological complete response rate (pCR) of 31% [32]. In this study, the presence of preexisting activated T-cells correlated with response. Patients with a high presence of intraepithelial CD8+ cells had a pCR rate of 40% (95% CI: 26–57%) compared to a rate of 20% (95% CI: 67–94%) for patients with an absence of CD8. A similar study called PURE-01 investigated neoadjuvant pembrolizumab in patients with MIBC and found enriched responses in patients with pre-existing CD8+ T-cell activation and a high TMB [33]. A recent analysis characterized the tumor and immune microenvironment incorporating digital spatial profiling in pre- and post-treatment tumors from the PURE01 to identify the histone demethylase KDM5B as a repressor of tumor-immune signaling pathways [34]. Moreover, in the resistant luminal-excluded subtype, the investigators demonstrated that inhibition of KDM5B enhances immunogenicity in FGFR3-mutated urothelial carcinoma cells. In contrast to neoadjuvant monotherapy with PD1/L1 inhibitors, the NABUCCO trial, which investigated neoadjuvant ipilimumab plus nivolumab in patients with MIBC, found that complete response was independent of baseline CD8+ presence or T-effector signatures [35]. However, they found that an induction of tertiary lymphoid structures (TLS) was observed in responding patients. A combination neoadjuvant immunotherapy was also evaluated in the DUTRENEO trial [36]. This trial assessed the efficacy of durvalumab and tremelimumab vs. chemotherapy as a neoadjuvant approach in patients with MIBC. Patients were classified as having a “hot” or “cold” tumor using a tumor inflammation signature score (TIS) based on an 18-gene interferon-y signaling related expression. Patients classified as having a hot tumor were randomized into the immunotherapy or chemotherapy groups while those classified as having cold tumors only received chemotherapy. This stratification failed to select patients more likely to benefit from immunotherapy vs. chemotherapy. These studies demonstrate that gene expression signatures might provide clues to the efficacy of select therapies, but further investigation is required.

## 4. FGFR and Other Receptor Tyrosine Kinase Inhibitors

Receptor tyrosine kinase (RTK) inhibitors have proven effective in cancer treatment, even achieving a 10-year survival rate of 83.3% in patients with chronic myeloid leukemia treated with imatinib. This has led to the potential of targeting this pathway in solid tumors, including UC [37]. In fact, the most identified clinically relevant genetic alterations in UC were cyclin-dependent kinase inhibitor 2A (CDKN2A, 34%), FGFR (21%), PIK3CA (20%), and ERBB2 (17%) [38]. Interestingly, the pan-FGFR inhibitor, erdafitinib, is currently the only FDA approved tyrosine kinase inhibitor (TKI) for post-platinum patients with locally advanced or mUC with susceptible FGFR3 or FGFR2 mutations or fusions [25]. The first study to compare FGFR-directed therapy with chemotherapy in patients with FGFR-over-expressing UC was the phase II FORT-1 trial, which showed comparable efficacy and safety [39]. Patients selected in this study were previously treated with platinum chemotherapy and had an overexpression of FGFR1 or FGFR3 mRNA. They were randomly assigned to treatment with rogaritinib vs. investigator-determined chemotherapy. An exploratory analysis suggested rogaritinib may have greater efficacy in patients with FGFR3 mRNA overexpression and an FGFR DNA-activating alteration, which may serve as biomarkers for prediction of RTK inhibitor response. However, the use of these requires further investigation, and potentially gene expression analysis, to identify FGFR pathway activity that may be superior to isolated analysis of FGFR1 or FGFR3.

The use of FGFR inhibitors as adjuvant therapy was being investigated in selected patients, but unfortunately had to close early due to poor accrual [40]. Pemigatinib is another FGFR1-3 inhibitor with promising data that is currently being evaluated in an open-label, single-arm, Phase II study as adjuvant therapy in high-risk UC patients post radical surgery (NCT04294277). Unfortunately, pan-FGFR inhibitors have shown increased toxicity compared to selective FGFR inhibitors [41]. As a result, the development of highly selective FGFR3 inhibitors is an area of interest in the treatment of urothelial carcinoma, and early Phase I trials of such agents are ongoing.

## 5. Antibody Drug Conjugates

As previously discussed, a ubiquitous challenge with cancer treatment is toxicity, which precision medicine has attempted to address. Antibody drug conjugates (ADCs) can selectively target tumor cells and spare the normal tissues, leading to a significant decrease in off-target side effects. The main components of an ADC are a monoclonal antibody (mAb), a linker, and a cytotoxic payload. Ideal antigenic targets are those that are highly expressed on malignant cells but not expressed in non-malignant cells [42]. Currently there are two FDA approved ADCs for treatment of UC, which are Enfortumab Vedotin-ejfv (EV) and Sacituzumab Govitecan-hziy (SG) [43,44]. Both EV and SG have been developed in unselected patients based on the nearly universal presence of their surface membrane targets, nectin4 and trop2, expressed by *NECTIN4* and *TROP2*, respectively. The phase II, single-arm EV-201 study of 125 patients with metastatic urothelial carcinoma, who previously received both a platinum-containing chemotherapy regimen and a PD-1/PD-L1 inhibitor, showed promising results in those who received EV with a confirmed ORR of 44% (95% CI, 35.1%–53.2%), including 12% CR [45]. Further, the EV-301 trial showed a statistically significant improvement in OS compared to the chemotherapy arm (median OS 12.88 months [95% CI: 10.58–15.21 months] vs. 8.97 months [95% CI: 8.05–10.74 months]) and superior PFS (median PFS of 5.55 months [95% CI: 5.32–5.82] vs. 3.71 months [95% CI: 3.52–3.94]). Moreover, EV was active as a second-line therapy in cisplatin-ineligible patients in a cohort of EV201, which led to approval in this context as well.

Similarly, TROPHY-U-01 demonstrated the beneficial use of SG in locally advanced or mUC that progressed after previous platinum-based chemotherapy and ICIs with an ORR of 27.4%, partial response (PR) of 22.1%, and a CR of 5.3% [44]. Phase III investigation is ongoing. A novel ADC called Disitamab Vedotin, also labeled as RC48-ADC, which targets HER2 and is comprised of hertuzumab coupled to monomethyl auristatin E (MMAE, a cell division inhibitor that blocks tubulin polymerization) via a cleavable linker, also showed ORR of 51.2% with an acceptable safety profile in patients with HER2+ locally advanced or mUC who were refractory to other therapies [46,47]. Further development is ongoing.

To predict EV sensitivity, Chu et al. retrospectively analyzed data by molecular subtyping and assessing levels of NECTIN4 expression from seven MIBC clinical cohorts (*n* = 1915) [48]. Although NECTIN4 expression was found to be heterogeneous across all molecular subtypes, it was significantly increased in luminal subtypes (Table 3). These data suggest that patients with the luminal subtype may have an increased sensitivity to EV. The authors also hypothesized a possible biomarker implication, as NECTIN4 expression was positively correlated with luminal markers *GATA3*, *FOXA1*, and *PPARG*. However, these biomarkers remain yet to be studied in the context of UC. An institutional retrospective study that investigated patients with advanced UC attempted to identify possible biomarkers in those who responded to EV [49]. The presence of *TP53* and absence of *CDKN2A* and *CDKN2B* alterations were associated with favorable responses and improved clinical outcomes. Another retrospective study using the data gathered from the much larger, multi-site UNITE study revealed other potential biomarkers [50]. Observed responses were higher in patients with *ERBB2* (67% vs. 44%; *p* = 0.05) and *TSC1* (68% vs. 25%; *p* = 0.04) alterations vs. wild-type. In addition, patients with high TMB were found to have longer median OS (13.5 vs. 8.3 months, *p* = 0.02). Shorter median PFS was found in patients with CDKN2A (4.4 vs. 6.0 months, *p* = 0.02), CDKN2B (4.4 vs. 6.0 months, *p* ≤ 0.01), and MTAP alterations (4.6 vs. 6.0 months, *p* = 0.05). Recently, a study suggested that metastatic tumors may exhibit lower NECTIN4 expression than the primary tumor, highlighting the potential value of proper tissue selection for biomarker analyses [51]. Further validation of these biomarkers is required before they can be clinically applied.

Biomarkers for SG sensitivity in the context of UC are currently being investigated. Chou et al. investigated expression levels of the drug target, TROP2, across different molecular subtypes of bladder cancer [52]. They found that TROP2 gene expression is higher across basal, luminal, and stroma-rich subtypes, but depleted in the neuroendocrine subtype, which may imply an increased sensitivity of certain tumor characteristics. Interestingly, prolonged exposure to EV may lead to downregulation of NECTIN4, which is associated with resistance to EV. However, these patients showed a response to SG without a change in Trop2 expression [52]. Indeed, SG appears to retain clinical activity even after prior EV. For this reason, the use of ADCs in combination or in sequence is an area of much anticipation [53].

## 6. Combination Therapy—Current Developments

Combination therapy has been an area of interest due to the genetic heterogeneity of UC and the variable response to known treatments. Interesting combinations that have been investigated include chemotherapy + PD1/PD-L1, FGFR inhibitor + PD1/PD-L1 inhibitor, Cytotoxic T-Lymphocyte Associated Protein 4 (CTLA-4) inhibitor + PD-1 inhibitor, and PD-L1 inhibitor + ADC. (Figure 2) (Refer to Appendix A for completed and ongoing studies for combination therapy of urothelial carcinoma).

IMvigor130 was the first study to report the combination of chemotherapy and ICI as a first-line treatment for advanced UC [54]. Patients were randomly assigned to receive atezolizumab plus platinum-based chemotherapy (group A), atezolizumab monotherapy (group B), or placebo plus platinum-based chemotherapy (group C). The addition of atezolizumab to chemotherapy resulted in a significant prolongation of progression-free survival (8.2 months [group A] vs. 6.3 months [group B]) and a near-doubling of complete responses (13% [group A] vs. 7% [group C]). There was an intriguing signal of greater benefit by combining atezolizumab with cisplatin-based as opposed to carboplatin-based chemotherapy. The recent analysis evaluating arms B and C demonstrated non-statistically significant OS benefit in the intention-to-treat analysis [55]. However, exploratory data showed clinical benefit with atezolizumab monotherapy in cisplatin-ineligible patients with PD-L1-high (IC2/3) tumors compared to placebo plus platinum-based chemotherapy (HR 0.56, 95% CI 0.34–0.91). KEYNOTE-361 is another randomized phase 3 trial that investigated the combination of chemotherapy and pembrolizumab as first-line treatment for advanced UC [56]. Unfortunately, this combination did not demonstrate a significant difference in PFS or OS. Other studies investigating the combination of chemotherapy + anti-PD1/PD-L1 are currently underway. CheckMate-901 incorporates a randomized phase II sub-study evaluating the combination of nivolumab with cisplatin plus gemcitabine, and results are eagerly awaited given the aforementioned signal of the potentially greater benefit of chemoimmunotherapy using a cisplatin-based backbone. A phase III trial is investigating atezolizumab with concurrent chemoradiation therapy (CRT) in patients with localized MIBC, with the primary outcome being bladder intact event-free survival (NCT03775265). The KEYNOTE-992 trial is investigating a similar hypothesis with pembrolizumab and CRT vs. CRT alone in patients with MIBC (NCT04241185).

FGFR inhibitors in combination with PD1/PD-L1 inhibitors are currently being studied in the FORT-2 and NORSE trials. FORT-2 (NCT03473756) is a phase Ib/II trial investigating rogaritinib and atezolizumab in patients with high FGFR1 or FGFR 3 expression and locally advanced or mUC. Preliminary data showed promising efficacy and safety in cisplatin-ineligible patients—most of whom were low or negative PD-L1 expression [57]. The NORSE trial (NCT03473743) is a phase Ib/II study investigating the combination of cetrelimab and erdafitinib in patients with mUC with specific FGFR alterations who progressed with ≥1 prior systemic therapy and had no prior FGFR therapy/PD-(L)1 inhibitors. Initial data from NORSE showed the combination of the two drugs has high antitumor activity vs. erdafitinib alone in mUC with an acceptable safety profile, and it is being further explored as a first-line treatment for patients ineligible for cisplatin [58]. While the combination of lenvatinib and pembrolizumab did not improve outcomes in a mostly platinum-ineligible first-line population (LEAP-011 phase III trial) [59], MAIN-CAV (NCT05092958) is another ongoing trial investigating cabozantinib (an anti-VEGFR TKI drug) and avelumab vs. maintenance avelumab alone following first-line platinum-based chemotherapy in patients with mUC.

The combination of nivolumab (PD-1 inhibitor) plus ipilimumab, a CTLA-4 inhibitor, has a demonstrated benefit in several tumor types. One of the cohorts in CheckMate-032 were patients with locally advanced or metastatic platinum-pretreated UC who were treated with nivolumab plus ipilimumab or nivolumab alone. ORR was reported to be 38% with nivolumab plus an ipilimumab dose of 3mg/kg compared to Nivolumab monotherapy with an ORR of 25.6% [60]. In the DANUBE Phase III trial, while the combination of durvalumab plus tremelimumab did not extend OS vs. gemcitabine-platinum, the subgroup of PD-L1-high patients appeared to have longer survival with durvalumab plus tremelimumab. However, the CheckMate-901 trial could not demonstrate improved survival for ipilimumab plus nivolumab vs. gemcitabine-platinum in PD-L1-high patients.

PD-L1 inhibitors have also been combined with ADCs. Promising activity for EV combined with pembrolizumab was seen in a nonrandomized phase Ib trial (EV-103 cohort A). The ORR was 73.3% with a complete response rate (CRR) of 15.6% and mOS of 26.1 months, suggesting potential synergism between these agents owing to immunogenic cell death [61]. EV-103 (NCT03288545) cohort K was a randomized phase II trial that compared EV alone versus its combination with pembrolizumab in patients with cisplatin-ineligible UC. The preliminary results were positive, with a higher ORR of 64.5% in the EV and pembrolizumab arm compared to 45.2% in the EV monotherapy arm, with tolerable safety profiles [62]. The median duration of response was ~22 months in the EV + pembrolizumab arm, and the results led to accelerated US FDA approval and suggest promise in the development of combination ADCs and ICIs. EV-302 (NCT04223856) is another ongoing Phase III trial investigating the combination therapy of EV and pembrolizumab versus chemotherapy alone in previously untreated locally advanced or mUC.

SG in combination with pembrolizumab is currently being investigated in perioperative patients with MIBC who cannot receive cisplatin-based chemotherapy in a phase 2 trial called SURE-02 (NCT03547973). SG is also being evaluated with other PDL-1 inhibitors including atezolizumab in an umbrella study (MORPHEUS-UC trial; NCT03869190). Another ongoing trial is investigating RC48-ADC (Disitamab Vedotin) in combination with toripalimab (a PD-1 inhibitor) in mUC (NCT04264936). The preliminary results revealed an ORR of 76.7% and a CR of 10%. The median PFS was immature, and the median OS was not reached. Interestingly, after subgrouping patients based on their level of HER2 expression and PD-L1 presence, they found that patients with the highest HER2 expression on immunohistochemistry (IHC 2+ or 3+) and PD-L1 positivity had the highest confirmed ORR (100%). In contrast, the ORR remained at 50% in subgroups with low HER2 (0 or 1+) regardless of expression of PD-L1 [63]. A larger phase III clinical trial is currently underway and is investigating RC48-ADC plus Toripalimab versus chemotherapy alone in previously untreated unresectable locally advanced or metastatic HER2-positive UC (NCT05302284). The JAVELIN bladder medley (NCT05327530) is an umbrella trial evaluating avelumab with one of three other antitumor agents including SG, M6223 (an anti-T-cell-immuno-receptor with Ig and ITM domains [anti-TIGIT]), or NKTR-255 (IL-15 receptor agonist that increases the proliferation and survival of NK cells) as a maintenance treatment in patients with locally advanced or mUC whose disease did not progress with first-line platinum-containing chemotherapy. One intriguing ongoing phase I trial (NCT04724018) is combining EV and SG to exploit their non-overlapping membrane targets, payloads, and toxicity profiles.

The combination of feladilimab and other anticancer agents, including pembrolizumab and chemotherapy, is currently being investigated in an umbrella study called INDUCE-1 (NCT04586244). Feladilimab is a humanized IgG4 antibody with activity against inducible T-cell Co-Stimulator (ICOS), which is a member of the CD28/B7/CTLA-4 receptor superfamily expressed on T-cells that augments T-cell proliferation, survival, and cytokine production. The preliminary data shows promising trends with an ORR of 8% vs. 22%, and disease control rate of 23% vs. 63% in feladilimab alone and feladilimab plus pembrolizumab, respectively [64]. This trial is currently underway and has shown encouraging results thus far.

A novel drug that is being evaluated in combination with pembrolizumab is soluble EphB4-human serum albumin (sEphB4-HSA). It is a recombinant fusion protein with antineoplastic and anti-angiogenic activities, composed of the full-length extracellular domain (soluble) of human receptor tyrosine kinase ephrin type-B receptor 4 (sEphB4) that is fused to full-length human serum albumin (HSA). Among patients whose tumors expressed EphrinB2 protein, the benefit appeared more robust, with a median OS of 21.5 months and ORR of 52%, including a CRR of 24% and median PFS of 5.7 months [65]. These findings exceeded the expectations across all end points and suggest that a combination of sEphB4-HSA and pembrolizumab can be another promising regimen in the therapeutic armamentarium against mUC. Further evaluation is ongoing (Table 4).

## 7. The Way Forward—Where Do We Go from Here?

Precision medicine in the context of UC has shown great potential with evidence of diagnostic and therapeutic value. The discovery of biomarkers and the development of risk stratification scores have given clinicians tools to help predict prognosis and therapeutic response. The identification of actionable mutations has shown promising results, with effective treatment options in patients with advanced disease. However, only a few of the identified mutations have had proven therapeutic value, and the search for effective targets continues. In this context, it is worth recalling that mTOR inhibitors have failed to demonstrate significant activity despite anecdotes of durable responses in the presence of genomic vulnerabilities, e.g., TSC1/2 alterations and mTOR-activating alterations [66,67,68]. PARP inhibitors have exhibited modest activity as single agents and in combination with immune checkpoint inhibition in those harboring somatic homologous recombination repair defects [69].

In this context, precision medicine may be even more important when developing therapies with promising activity, coupled with potentially life-threatening toxicities, in order to improve the therapeutic index and cost efficacy. For example, adoptive T-cell therapy (ACT) has demonstrated efficacy in treating hematological malignancies. However, despite its prominent success in treating hematological cancers, it has been difficult to obtain a similar rate of success in solid tumors, including urothelial cancers, due to lack of specificity of antigens, low affinity of T-cell receptors (TCRs) for most tumor antigens, and slow migration and enrichment rate of CAR-T-cells into solid tumors, in addition to the immunosuppressive nature of the tumor microenvironment (TME). Increasing the dose of CAR-T-cells is a possible solution, but this can induce higher toxicity due to off-target effects [70]. There are multiple studies investigating the use of CAR-T-cell therapy in various solid tumors, including UC. A phase 1 study is investigating HER2-specific CAR-T-cells in combination with an intra-tumor injection of CAdVEC, which is an oncolytic adenovirus designed to increase reactivity and the efficacy of treatment (NCT03740256). Another study is evaluating CCT3-1-59 T-cells in patients with receptor tyrosine kinase-like orphan receptor 2 (ROR2) who have stage IV metastatic solid tumors (NCT03960060). A phase I/II study is investigating patients with locally advanced or mUC with 4SCAR-T-cells that are specifically targeted against prostate-specific membrane antigen (PSMA) and folate receptor alpha (FRa) (NCT03185468). These ongoing studies aim to improve treatment efficacy by providing targets that increase the accuracy and reactivity of CAR-T-cell therapy.

Given the variable response of current treatment modalities, a combination therapy of known targets is also being studied. However, the effectiveness of this approach still requires further investigation. The application of ACT in UC continues to be studied. As discoveries are made, therapeutic modalities endeavor to increase efficacy while decreasing toxicity. Many breakthroughs, including increasingly precise diagnostic tools and therapies, have laid the foundation and paved the way for future innovations and confirmatory studies. Clinicians currently have some tools that allow them to take advantage of individual patient and tumor characteristics to treat UC. However, as outlined above, there is still much room for improving treatment efficacy and predicting prognosis and treatment response. Moreover, the prediction of severe toxicities needs greater focus to optimize the therapeutic index of anti-cancer therapy.

## Figures and Tables

**Figure 1 cancers-15-03024-f001:**
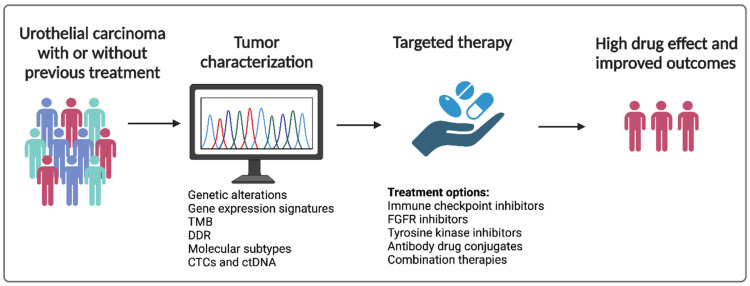
Precision medicine in urothelial carcinoma. TMB: tumor mutational burden; DDR: DNA damage response; CTC: circulating tumor cell; ctDNA: circulating tumor DNA; and FGFR: fibroblast growth factor receptor. Adapted from “Precision Cancer Therapy” by BioRender.com (2023). Retrieved from https://app.biorender.com/biorender-templates (accessed on 28 March 2023).

**Figure 2 cancers-15-03024-f002:**
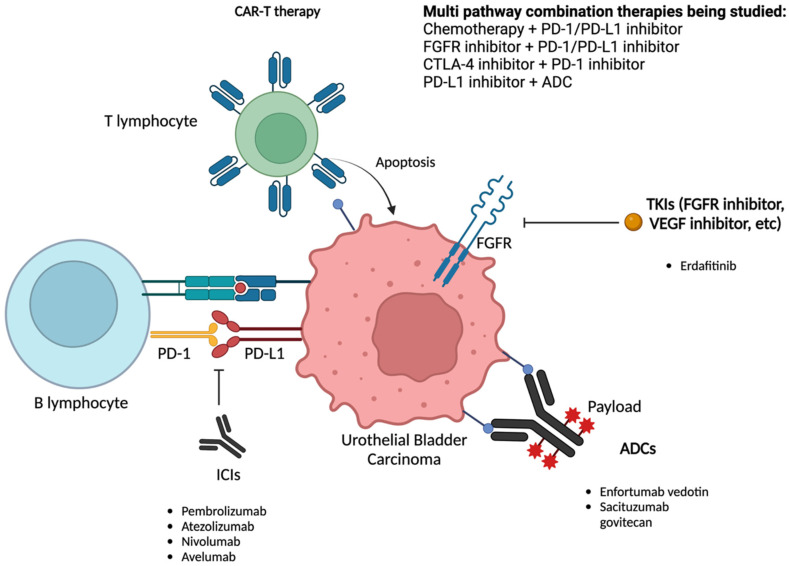
Current targeted therapies for urothelial carcinoma. CAR-T: chimeric antigen receptor therapy; PD-1: programmed cell death protein 1; PD-L1: programmed death-ligand 1; CTLA 4: cytotoxic T-lymphocyte-associated protein 4; ADC: antibody drug conjugate; ICI: immune checkpoint inhibitor; FGFR: fibroblast growth factor receptor; and VEGF: vascular endothelial growth factor. Created with BioRender.com (accessed on 28 March 2023).

**Table 1 cancers-15-03024-t001:** Biomarkers for prediction of ICI response in metastatic or locally advanced urothelial carcinoma.

Study	Study Population	Treatment Arms	Biomarker	Improved Outcomes
Phase III IMvigor211 [21]	locally advanced or mUC after progression with platinum-based chemotherapy	atezolizumab vs. chemotherapy	TMB	median OS was longer with atezolizumab (HR 0.68, CI 95% 0.51–0.90) in the TMB-high subgroup
Phase II Checkmate 275 [16]	metastatic or surgically unresectable locally advanced UC	nivolumab	TMB	TMB alone showed a positive association with ORR with nivolumab [OR (95% CI): 2.13 (1.26–3.60), *p* < 0.05]
			ARID1A + CXCL13	ARID1A mutation and high CXCL13 was associated with median PFS of 3.7 months (85% CI, 1.8 to NA) and OS of 19.1 months (95% CI, 6.1 to NA) compared to patients with neither alteration who had a median PFS of 1.9 months (95% CI, 1.7–2.0) and median OS of 5.3 months (95% CI, 3.6 to 11.4)
			molecular subtypes (luminal 1, luminal 2, basal 1, basal 2)	Basal 1 (cluster 1) and luminal 2 (cluster 2) were found to have higher CR rates to nivolumab compared to the other subtypes (Basal 2 CR, 0%; luminal 1 CR, 1.5%; luminal 2 CR, 1.8%).
			High interferon gamma signature	more likely to respond to nivolumab than those with low expression (*p* = 0.0003)
Nassar et al.—A model combining clinical and genomic factors to predict response to PD-1/PD-L1 blockade in advanced urothelial carcinoma [24]	mUC	Retrospective study of ICIs	TMB/SNV count, visceral metastases and NLR highly	Biomarkers highly correlated with tumor regression [AUC (95% CI) = 0.90 (0.80, 0.99)] [24]
Phase II IMvigor210 [28]	Locally advanced or mUC	Atezolizumab	ARID1A + CXCL13	expression of both ARID1A mutation plus baseline CXCL13 expression had a median OS of 17.8 months (95% CI, 10.4 to NA) compared to 7.1 months (95% CI, 5.5 to 9.9) in patients with neither

Legend: mUC: metastatic urothelial carcinoma; TMB: tumor mutational burden; OS: overall survival; HR: hazard ratio; CI: confidence interval; ORR: overall response rate; OR: odds ratio; PFS: progression free survival; CR: complete response; AUC: area under the curve; SNV: single nucleotide variants; neutrophil-lymphocyte ratio; PD-1: programmed cell death-1; and PD-L1: programmed cell death ligand-1.

**Table 2 cancers-15-03024-t002:** Biomarkers for the prediction of ICI Response for neoadjuvant therapy in patients with MIBC.

Study	Cancer Type	Treatment Arms	Biomarker	Outcomes
ABACUS Study [32]	MIBC	neoadjuvant atezolizumab	preexisting activated T-cells	Patients with a high presence of intraepithelial CD8+ cells had a pCR rate of 40% (95% CI: 26–57%) compared to a rate of 20% (95% CI: 67–94%) for patients with an absence of CD
PURE-01 [33,34]	MIBC	neoadjuvant pembrolizumab	histone demethylase KDM5B	inhibition of KDM5B enhances immunogenicity in FGFR3-mutated UC cells
NABUCCO trial [35]	MIBC	neoadjuvant ipilimumab plus nivolumab	Tertiary lymphoid structures	induction of TLS was observed in responding patients
DUTRENEO trial [36]	MIBC	Neoadjuvant durvalumab and tremelimumab vs. chemotherapy	Patients were classified as having a “hot” or “cold” tumor using a TIS score based on an 18-gene interferon-y signaling related expression	This stratification failed to select patients more likely to benefit from immunotherapy vs. chemotherapy.

Legend: MIBC: muscle-invasive bladder cancer; pCR: pathologic complete response; CI: confidence interval; UC: urothelial carcinoma; TLS: tertiary lymphoid structures; TIS: tumor inflammation signature.

**Table 3 cancers-15-03024-t003:** Biomarkers to predict EV response.

Study	Cancer Type	Study Design	Biomarker	Conclusions/Outcomes
Chu et al.—Heterogeneity in *NECTIN4* Expression Across Molecular Subtypes of UC Mediates Sensitivity to EV [48]	MIBC	Retrospective study with molecular subtyping and NECTIN4 expression data from seven MIBC clinical cohorts	Molecular subtyping and NECTIN4 expression	Sensitivity to EV is mediated by expression of *NECTIN4*, which is enriched in luminal subtypes of bladder cancer
Jindal et al.—Biomarkers predictive of response to EV treatment in advanced UC [49]	Advanced UC	Retrospective study assessing molecular and clinical characteristics	*TP53*, *CDKN2A*, *CDKN2B*	The presence of *TP53* and absence of *CDKN2A* and *CDKN2B* alterations were associated with favorable responses and improved clinical outcomes.
UNITE study [50]	Advanced UC	Retrospective study assessing molecular and clinical characteristics	*ERBB2*, *TSC1*	Observed responses were higher in patients with *ERBB2* (67% vs. 44%; *p* = 0.05) and *TSC1* (68% vs. 25%; *p* = 0.04) alterations vs. wild-type.
			High TMB	patients with high TMB were found to have longer median OS (13.5 vs. 8.3 months, *p* = 0.02)
			CDKN2A, CDKN2B, MTAP alterations	Shorter median PFS was found in patients with CDKN2A (4.4 vs. 6.0 months, *p* = 0.02), CDKN2B (4.4 vs. 6.0 months, *p* ≤ 0.01), and MTAP alterations (4.6 vs. 6.0 months, *p* = 0.05).

Legend: UC: urothelial carcinoma; EV: enfortumab vedotin; MIBC: muscle-invasive bladder cancer; TMB: tumor mutational burden; OS: overall survival; PFS: progression free survival.

**Table 4 cancers-15-03024-t004:** Currently Ongoing Clinical Trials Pending Results for Targeted Therapies for Urothelial Carcinoma (Combination therapy).

Name(s)	Target/MOA	Trials	Phase	*n*	Study Arms	Primary Endpoint(s)
Atezolizumab + CRT	PD-L1 + CRT	NCT03775265	III, active, recruiting	475	RT + chemotherapy vs. CRT + atezolizumab in localized MIBC	Bladder intact event-free survival
Pembrolizumab + CRT	PD-1 + CRT	NCT04241185 (KEYNOTE-992)	III, active, recruiting	636	Pembrolizumab + CRT vs. placebo + CRT in MIBC	Bladder intact event-free survival
Enfortumab vedotin + pembrolizumab + platinum-based chemotherapy	Nectin-4 and monomethyl auristatin E (MMAE) + PD-1	NCT03288545 (EV-103)	I/II, active, not recruiting	457	UC	ORR, pCR, AEs
Saciuzumab-govitecan + pembrolizumab	Anti-Trop-2 humanized monoclonal antibody + PD-1/PD-L1	NCT05535218 (SURE-02)	II, active, recruiting	48	MIBC	pCR
Sacituzumab-govitecan + Atezolizumab		NCT03869190 (MORPHEUS-UC)	Ib/II, active, recruiting	645	MIBC or locally advanced or mUC who progressed with platinum therapy	ORR, pCR
Sacituzumab-govitecan + Avelumab		NCT05327530 (JAVELIN Bladder Medley)	II, active, recruiting	252	MIBC or locally advanced or mUC who progressed with platinum therapy	PFS, AEs
Disitamab Vedotin + Pembrolizumab	HER2 (hertuzumab and MMAE) + PD-1/PD-L1	NCT04879329	II, active	270	Disitamab Vedotin monotherapy (only cohort C) for HER2+ locally advanced unresectable or mUC	Confirmed ORR
Disitamab Vedotin + Toripalimab		NCT05302284	III, active, recruiting	456	untreated unresectable locally advanced or metastatic HER2-positive UC	PFS, OS
RC48-ADC (Disitamab Vedotin) + Toripalimab		NCT04264936	Ib/II, active, unknown recruitment status	36	RC48-ACD and JS001 for locally advanced or mUC	AEs and maximal tolerated dose
EphB4-human serum albumin + pembrolizumab	EphB4-human serum albumin + PD-1	NCT02717156	II, active, recruiting	170	EphB4-HAS + pembrolizumab in solid tumors	Toxicities and AEs
Cabozantinib + avelumab (VEGF TKI + PD-L1 inhibitor)	VEGF TKI + PD-L1 inhibitor	NCT05092958 (MAIN-CAV)	III, active, recruiting	654	Avelumab vs. avelumab + cabozantinib in mUC	OS

Legend: MOA: mechanism of action; ORR: Objective Response Rate; PFS: Progression Free Survival; OS: Overall Survival; AE: adverse reaction; COR: Confirmed Objective Response; DFS: Disease Free Survival; DLT: dose-limiting toxicity; TEAE: Treatment-emergent adverse events; BOR: Best Objective Response Rate; UC: urothelial carcinoma; ADC: Antibody-Drug Conjugate; mUC: metastatic urothelial carcinoma; CRT: chemoradiotherapy; pCR: pathological complete response; MIBC: Muscle-invasive bladder cancer; and AE: Adverse events.

## Data Availability

No new data were created or analyzed in this study. Data sharing is not applicable to this article.

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
