# Peer review of "Precision Medicine to Treat Urothelial Carcinoma—The Way Forward"

_cancers, 2023, doi:10.3390/cancers15113024_

Round 1

Reviewer 1 Report

The review presented by Luceno et al. is a thorough summary of recent activities in precision medicine for urothelial carcinoma. 

The only consideration I have concerns the fact that the review is a detailed enumeration of facts, leaving most of the urologists and translation oriented cancer researchers who want to learn, still uncertain. The paper does not - or too little - accentuate the different aspects of potential treatment areas, and mostly relates to studies of the metastasized tumor. I would suggest to either indicate this in the introduction or structure the options according to the tumor with metastasis, the analysis of non metastasized local tumor features for treatment strategies, the treatment of Carcinoma in situ after BCG failure, neoadjuvant therapy, adjuvant therapy, assessment of therapy response prior to therapy vs. therapy response during therapy in general.  This structure would help the readers who want to orient their research clinically and would help the reader who is a urooncologists to easier categorize  and evaluate the information provided.

Author Response

Thank you for your review. 

For clarity, we included the following sentence at the end of the second paragraph of the introduction: "We explore how biomarkers may imply favorable treatment outcomes for patients with MIBC and locally advanced or metastatic UC. The possibility of selecting patients who should receive neoadjuvant or adjuvant therapies according to liquid biopsies, prognostic tools, and biomarkers is also discussed."

We believe that this will offer transparency in what the review offers. We also added charts to clarify which biomarkers were studied in their specific cancer type. For example, Table 1 is titled "Biomarkers for prediction of ICI response in metastatic or locally advanced urothelial carcinoma" and Table 2 is titled Biomarkers for the prediction of ICI Response for neoadjuvant therapy in patients with MIBC. There is a 3rd table that outlines the biomarkers that may help predict EV response with the respective cancer type identified.

These changes give context to each biomarker and their respective  cancer stage according to each study. We hope this assists readers digest and evaluate the information provided in this review. 

Reviewer 2 Report

This review is about the treatment of urothelial carcinoma (UC) and variable response to current therapies. To address this, many tools, including tumor biomarker assessment and liquid biopsies have been developed to predict prognosis and treatment response. The authors highlight advancements in the treatment of UC, describe ongoing clinical trials, and identify areas for future study in the context of precision medicine. 

I suggest for clarity that all clinical trials should be presented as a table, describing details of groups, and the clinical phase of trials. 

Author Response

Thank you for your review. 

We added 4 Tables to our manuscript that highlights the pertinent studies in this review, which are titled as follows: 

Table 1. Biomarkers for prediction of ICI response in metastatic or locally advanced urothelial carcinoma

Table 2. Biomarkers for the prediction of ICI Response for neoadjuvant therapy in patients with MIBC

Table 3. Biomarkers to predict EV response

Table 4. Currently Ongoing Clinical Trials Pending Results for Targeted Therapies for Urothelial Carcinoma (Combination therapy)

These tables identify the details of each group studied and the clinical phase of the trials.